# Anti-Allergic and Anti-Inflammatory Effects of Lidocaine-Derived Organic Compounds in a House Dust Mite-Induced Allergic Rhinitis Mouse Model

**DOI:** 10.3390/biomedicines12091965

**Published:** 2024-08-29

**Authors:** Seung-Heon Shin, Mi-Kyung Ye, Mi-Hyun Chae, Sang-Yen Geum, Ahmed S. Aboraia, Abu-Baker M. Abdel-Aal, Wesam S. Qayed, Hend A. A. Abd El-wahab, Ola F. Abou-Ghadir, Tarek Aboul-Fadl

**Affiliations:** 1Department of Otolaryngology-Head and Neck Surgery, School of Medicine, Daegu Catholic University, Daegu 42472, Republic of Korea; miky@cu.ac.kr (M.-K.Y.); leonen@hanmail.net (M.-H.C.); saye60@naver.com (S.-Y.G.); 2Department of Medicinal Chemistry, Faculty of Pharmacy, Assiut University, Assiut 71526, Egypt; ahmed.mohamed15@pharm.aun.edu.eg (A.S.A.); wesam.qayed@aun.edu.eg (W.S.Q.); hendaboelmaged@aun.edu.eg (H.A.A.A.E.-w.); fadl@aun.edu.eg (T.A.-F.); 3Department of Pharmaceutical Organic Chemistry, Faculty of Pharmacy, Assiut University, Assiut 71526, Egypt; abubaker.elsayed@pharm.aun.edu.eg (A.-B.M.A.-A.); olaghadir@aun.edu.eg (O.F.A.-G.)

**Keywords:** allergic rhinitis, mouse model, lidocaine analog, inflammation, cytokine

## Abstract

Allergic rhinitis (AR) is a common chronic disease that significantly impacts the quality of life. Lidocaine is known to have anti-inflammatory and immunomodulatory effects. This study evaluated the effect of lidocaine analogs in a *Dermatophagoides pteronyssinus* (DP)-induced AR mouse model. An AR model was developed using BALB/c mice via intraperitoneal sensitization with DP and intranasal challenge with DP. One hour before stimulation with DP, lidocaine analogs, EI137 and EI341 (at a dose of 0.5 or 5 ug/g), were administered intranasally. Nasal symptoms and serum total IgE, interleukin (IL)-4, IL-10, interferon (IFN)-γ, and tumor necrosis factor (TNF)-α levels were evaluated. Reverse-transcription polymerase chain reaction was used to determine IL-4, IL-10, and IFN-γ, as well as the expression of their mRNA transcription factors in the sinonasal mucosa. Histologic changes were evaluated using hematoxylin and eosin and periodic acid–Schiff staining. The DP-induced AR mouse model had increased serum levels of total IgE and cytokines. EI137 and EI341 significantly suppressed the levels of total IgE, IL-4, and TNF-α. Intranasal instillation of EI137 and EI341 significantly inhibited IL-4, IL-10, and IFN-γ mRNA expression, as well as inflammatory cells and mucus-producing goblet cells. Lidocaine analogs also suppressed DP-stimulated IL-4, IFN-γ, and IFN-γ production by splenocytes. Intranasal instillation of EI137 and EI341 exhibited anti-allergic and anti-inflammatory effects, influenced by Th1 and Th2 inflammatory cytokines. These lidocaine analogs suppressed DP-induced sinonasal mucosal inflammation, inflammatory cell infiltration, and mucus hypersecretion.

## 1. Introduction

Allergic rhinitis (AR) is one of the most common chronic diseases; it is an IgE-mediated nasal mucosal inflammatory disease with a significant impact on the quality of life and a huge socioeconomic burden [1]. The median worldwide prevalence of AR is 18.1%, which has increased over time due to global warming, lifestyle changes, and air pollution [2]. Traditional treatments, including allergic avoidance and pharmacotherapy with antihistamines, corticosteroids, leukotriene modifiers, and immunotherapy, often fail to provide satisfactory symptom control [1,3]. Recent advances in understanding the pathogenesis of AR highlight the potential of biologics, though their clinical use is limited by cost and side effects [3,4]. 

Lidocaine has long been used clinically as a local anesthetic and anti-arrhythmic agent; additionally, it also exhibits anti-inflammatory and immunomodulatory effects [5,6]. An in vitro study revealed that lidocaine induces eosinophil apoptosis and inhibits cytokine-induced superoxide production [7]; however, topical inhalation of lidocaine can induce transient oropharyngeal anesthesia. It also has a bitter taste, and aerosolized lidocaine itself can trigger bronchoconstriction and hyperreactivity in asthmatic patients, thus limiting its clinical use [8]. To overcome these limitations, several studies have investigated lidocaine derivatives [9,10]. These derivatives exhibit anti-spasmodic, anti-inflammatory, anti-eosinophilic activity, and anti-bronchoconstrictive effects without the local anesthetic properties of lidocaine; however, these lidocaine derivatives are not yet commonly used clinically.

Our previous in vitro study demonstrated that among 36 lidocaine-derived organic compounds, EI137 and EI341 significantly inhibit interleukin (IL)-5-mediated eosinophil activation, survival, and transcription factor expression [10]. Additionally, they effectively suppress the production of IL-8 and eosinophil cationic protein from eosinophils. We hypothesized that if EI137 and EI341 possess anti-eosinophilic properties, then they could influence the development or treatment of upper airway allergic inflammatory diseases. To confirm the anti-inflammatory and anti-allergic effects of EI137 and EI341 in a house dust mite-induced AR mouse model, we conducted a study analyzing the inflammatory responses, chemical mediators, and transcription factors in the sinonasal mucosa.

## 2. Materials and Methods

### 2.1. Reagents

*Dermatophagoides pteronyssinus* (DP) was purchased from Greer Lab (Lot No 410262, Lenoir, NC, USA). Phosphate-buffered saline (PBS), penicillin, and streptomycin were obtained from Gibco (Grand Island, NY, USA). Lidocaine was obtained from Sigma-Aldrich (St. Louis, MO, USA). Aluminum hydroxide (21645-51-2), Trizol (15596018), and mouse IgE-uncoated enzyme-linked immunosorbent assay (ELISA) kit (88-50460-88) were purchased from Invitrogen (Carlsbad, CA, USA). ELISA kits for interleukin (IL)-4 (M4000B-1), IL-10 (M1000B-1), interferon (IFN)-γ (MIF00-1), and tumor necrosis factor (TNF)-α (MTA00B-1) were purchased from R&D system (Minneapolis, MN, USA). SYBR Green PCR core kit was purchased from PE Applied Biosystems (Foster City, CA, USA). RBC lysis buffer was acquired from BioLegend (San Diego, CA, USA).

### 2.2. Preparation of Lidocaine-Derived Organic Compounds

EI137 and EI341 were synthesized according to the methods reported by the Department of Pharmaceutical Organic and Medicinal Chemistry, Assiut University, Assiut, Egypt (Figure 1) [11]. The aromatic ring and acyl moiety of lidocaine were modified such that the resulting compounds lacked the tertiary amine responsible for sodium-channel-blocking activities. Their structures and purity were verified based on spectral and elemental methods of analysis as previously reported [11]. The molecular weight of both EI137 and EI341 was 259.7 g/mol.

### 2.3. Development of DP-Induced Allergic Rhinitis Mouse Model and Experimental Protocol

Six-week-old female BALB/c mice free of murine-specific pathogens were obtained from Hyosung Science Inc. (Daegu, Republic of Korea). They were kept under specific pathogen-free conditions in standard laboratory cages with free access to food and water. All animal experiments in this study were conducted in accordance with the guidelines of the National Institute of Health and approved by the Institutional Review Board of Animal Experiments of Daegu Catholic University Medical Center.

The mice were sensitized by intraperitoneal injection with 100 μg of DP and 2 mg aluminum hydroxide in 200 μL of PBS on days 0, 7, and 14. From days 15 to 21 after initial sensitization, 10 μg of DP in 20 μL of PBS was administered intranasally in both nostrils. From days 22 to 28, 0.5 μg/g or 5 μg/g of EI134 or EI341 in 20 μL of PBS was administered intranasally in both nostrils 1 h before DP stimulation. The mice were randomly divided into seven groups (n = 8 each). Group 1, the negative control mice, were stimulated with PBS instead of DP. Group 2 received PBS instead of organic compounds and was then stimulated with DP. Group 3 was treated with 5 μg/g of lidocaine prior to DP stimulation. Groups 4 and 5 were treated with 0.5 and 5 μg/g of EI134, respectively. Groups 6 and 7 were treated with 0.5 and 5 μg/g of EI347, respectively (Appendix A). Since intranasal instillation of lidocaine is known to have anti-inflammatory and anti-allergic effects in a murine model of allergic rhinitis, we used lidocaine as a positive control [12].

### 2.4. Evaluation of Allergic Symptoms

Mice were placed in an observation cage for approximately 10 min for acclimatization. Then, 10 μg of DP was administrated intranasally, and nasal symptoms were evaluated 5 min later by counting the number of nasal rubbings and the number of sneezes over a 15 min period following the last stimulation. Results between groups were compared.

### 2.5. Evaluation of Inflammatory Cells in Nasal Lavage Fluid (NLF)

NLF was collected 24 h after the last intranasal stimulation with DP. A 20-gauge catheter was inserted in the direction of the nasopharynx through a partially resected trachea. The nasal cavity was gently irrigated with 1 mL of cold PBS and the extracted NLF was centrifuged at 2000 rpm for 7 min at 4 °C. The pellet was then resuspended in PBS, and 10 μL of the cell suspension was stained with May–Grunwald–Giemsa stain. Cells were differentiated into eosinophils, neutrophils, lymphocytes, and other cells, and the average number of cells was determined based on five high-power field views.

### 2.6. Measurement of Serum Total IgE and Cytokines

Peripheral blood was collected from the inferior vena cava 24 h after the last intranasal provocation. Serum was obtained by centrifugation and stored at −70 °C. Total IgE, IL-4, IL-10, IFN-γ, and TNF-α levels were determined using commercially available ELISA kit. The limit of detection was 4 ng/mL for IgE and less than 2 pg/mL for each cytokine.

### 2.7. Measurement of Cytokines and Transcription Factor mRNA in Sinonasal Mucosa

Total RNA was extracted from the sinonasal mucosa using the Trizol reagent (Invitrogen). RNA purity and concentration were measured using a spectrophotometer (FlUOstar Optima, BMG Labtech, Ortenberg, Germany). Complementary DNA was generated from 1 μg of RNA using reverse-transcription polymerase chain reaction (RT-PCR) amplification with a Bio-Rad (CFXOpus, Hercules, CA, USA) thermal cycler. Quantitative PCR was then performed using a SYBR Green PCR core kit and the amplified cDNA. The primer sequences and amplified products were as follows: IL-4 sense 5′-CAA TTG CAA TGC CAT CTA CAG GAC-3′ and antisense 5′-TTT TGG TAT CGG GGA GGC TG-3′ (104 bp), IL-10 sense 5′-GCC AGA GCC ACA TGC TCC TA-3′ and antisense 5′-GAT AAG GCT TGG CAA CCC AAG TAA-3′ (145 bp), IFN-γ sense 5′-CGG CAC AGT CAT TGA AAG CCT A-3′ and antisense 5′-GTT GCT GAT GGC CTG ATT GTC-3′ (199 bp), T-bet sense 5′-GCC AGG GAA CCG CTT ATA-3′ and antisense 5′-CCT TGT TGT TGG TGA GCT TTA-3′ (104 bp), GATA-3 sense 5′-TAC CAC CTA TCC GCC CTA TG-3′ and antisense 5′-GCC TCG ACT TAC ATC CGA AC-3′ (101 bp), Foxp3 sense 5′-CAC CTA TGC CAC CCT TAT CCG-3′ and antisense 5′-CAT GCG AGT AAA CCA ATG GTA GA-3′ (91 bp), and β-actin sense 5′-GCA GAA GGA GAT TAC TGC TCT-3′ and antisense 5′-GCT GAT CCA CAT CTG CTG GAA-3′ (136 bp). Initial denaturation was performed at 95 °C for 2 min, followed by 40 cycles consisting of denaturation at 94 °C for 10 s, annealing at 60 °C for 10 s, and elongation at 72 °C for 45 s. Two technical replicates and 3–4 biological replicates were prepared for mRNA studies. The expression levels of mRNA were measured using the cycle threshold (2^−ΔΔCT^) method and normalized to β-actin.

### 2.8. Measurement of Cytokines from Splenocytes Stimulated with DP

Splenocytes were isolated from the spleen tissues of each group by grounding and separating them into single cells using a 70 μm cell strainer. Red blood cells (RBCs) were removed using RBC lysis buffer. The splenocytes were then incubated in Roswell Park Memorial Institute 1640 medium supplemented with 10% fetal bovine serum, 100 U/mL of penicillin, and 100 μg/mL of streptomycin. A total of 5 × 10^6^ splenocytes were further incubated with 10 μg/mL of DP for 72 h. The supernatant was collected, and IL-4, IL-10, IFN-γ, and TNF-α levels were measured using ELISA quantitation kits.

### 2.9. Histological Evaluation of Sinonasal Mucosa

Mice were painlessly euthanized with a lethal dose of intraperitoneal pentobarbital sodium 24 h after the last intranasal provocation with DP. The specimens were decalcified in ethylenediaminetetraacetic acid and embedded in paraffin. Tissue sections were cut anteroposteriorly into 5-μm-thick coronal sections. Three anatomically similar sections were selected from each mouse, as described in a previous study [13]. Inflammatory cell infiltration and epithelial thickness were quantified using hematoxylin-and-eosin-stained sections at ×400 magnification. Goblet cell numbers were quantified using periodic acid Schiff staining at ×200 magnification. The degree of submucosal inflammatory cell infiltration was quantified into four categories ranging from 0 to 3 (0: none, 1: mild occasional scattered inflammatory cells, 2: moderate, 3: severe diffuse infiltration of inflammatory cells). Epithelial thickness was directly measured using a video camera (Olympus Optical Co., Ltd., Tokyo, Japan) and analyzed with DP controller software (ver.2.2.1.227). Goblet cells were counted using an eyepiece reticle. All tissue sections were examined blindly with respect to tissue source, and the mean counts were determined at three different mucosal areas for each of the three sections per mouse.

### 2.10. Statistical Analysis

All measured parameters are expressed as the mean ± standard deviation of eight representative independent experiments for each group. Student’s *t*-test was used for pairwise comparisons of two groups, whereas comparisons between several groups utilized one-way analysis of variance followed by Tukey’s test for normally distributed data. For non-normally distributed data, the Mann–Whitney U test and the Kruskal–Wallis test with post hoc Bonferroni–Dunn test were performed using SPSS ver. 21 software (IBM Corp., Armonk, NY, USA). Statistical significance was set at *p* < 0.05.

## 3. Results

### 3.1. Effects of Lidocaine Analogs on the Allergy Symptoms

Allergic nasal symptoms, including the frequency of sneezing and nasal rubbing, were counted 15 min after the last challenge with DP. In the DP-challenged group, sneezing and nasal rubbing were significantly more frequent than in PBS-stimulated negative control mice. Sneezing and nasal rubbing due to DP challenge were dose-dependently inhibited by intranasal instillation of EI137 and EI341 at doses of both 0.5 and 5 μg/g. Additionally, intranasal instillation of 5 μg/g of lidocaine also significantly inhibited the frequency of sneezing and nasal rubbing (Figure 2).

### 3.2. Effect of Lidocaine Analogs on Inflammatory Cell Differentiation in NLF

Eosinophil and neutrophil counts in the NLF were significantly increased in NLF (eosinophil, 39.3 ± 12.7; neutrophil, 24.3 ± 10.8) when sensitized mice were challenged with DP compared to nonallergic negative control mice (eosinophil, 1.6 ± 0.6; neutrophil, 3.2 ± 1.5). Eosinophil and neutrophil counts in the NLF were significantly decreased in a dose-dependent manner by intranasal instillation of lidocaine analogs, EI137 and EI341 (0.5 μg/g of EI137, 13.0 ± 9.3 and 13.2 7.2, respectively; 5 μg/g of EI137, 3.5 ± 1.2 and 8.0 ± 6.4, respectively; 0.5 μg/g of EI341, 24.0 ± 12.9 and 15.3 ± 7.9, respectively; and 5 μg/g of EI341, 7.3 ± 3.6 and 5.5 ± 1.7, respectively). Intranasal instillation of 5 μg/g of lidocaine also significantly inhibited the number of these inflammatory cells in NLF (Figure 3). Although a few lymphocytes were observed in the NLF, there were no differences between the experimental groups.

### 3.3. Effect of Lidocaine Analogs on Serum Total IgE and Cytokine Levels

Allergic inflammatory responses were assessed by measuring serum total IgE, IL-4, IL-10, IFN-γ, and TNF-α levels using ELISA. DP-challenged mice had significantly increased IgE levels compared to the nonallergic negative control mice. These IgE levels, which were increased by DP, were significantly decreased by intranasal instillation of EI137 and EI137 at doses of both 0.5 μg/g and 5 μg/g; however, IgE levels were not significantly affected by intranasal instillation of 5 μg/g of lidocaine (Figure 4A).

The DP-challenged mice had significantly increased IL-4, IL-10, IFN-γ, and TNF-α levels compared to the nonallergic negative control mice. Intranasal instillation of EI137 at doses of both 0.5 μg/g and 5 μg/g significantly suppressed IL-4 and TNF-α levels, while 5 μg/g of EI341 significantly suppressed IL-4, IL-10, and TNF-α levels compared to untreated mice. Intranasal instillation of lidocaine analogs did not inhibit serum IFN-γ levels. The Intranasal instillation of 5 μg/g of lidocaine only significantly influenced TNF-α levels (Figure 4B,C).

### 3.4. Effect of Lidocaine Analogs on Cytokines and the Expression of Their Transcription Factor mRNA in Sinonasal Mucosa

Real-time RT-PCR was conducted to assess the effect of intranasal instillation of lidocaine analogs, EI137 and EI341, on the mRNA expression of Th-related cytokine and T-cell subset transcription factors. Intranasal challenge with DP significantly elevated the IFN-γ, IL-4, and IL-10, and Th2 transcription factors GATA-3 mRNA expression in the sinonasal mucosa. The increased mRNA expression of IFN-γ, IL-4, IL-10, and GATA-3 was significantly suppressed by intranasal instillation of EI137 and EI341 at doses of both 0.5 μg/g and 5 μg/g, as well as by 5 μg/g of lidocaine compared to untreated mice (Figure 5A,B).

### 3.5. Effect of Lidocaine Analogs on DP-Induced Splenocyte Activation

The systemic effects of lidocaine analogs, EI137 and EI341, were evaluated using splenocytes. Splenocytes isolated from each experimental group were stimulated with 10 μg/mL of DP, and IL-4, IL-10, IFN-γ, and TNF-α levels were determined using ELISA. Splenocytes isolated from the DP-induced allergic rhinitis mouse model exhibited significantly increased IL-4, IL-10, IFN-γ, and TNF-α levels after stimulation with DP. Splenocytes isolated from mice treated with intranasal instillation of EI137 (5 μg/g) and EI341 (0.5 and 5 μg/g) showed a significant decrease in IL-4, IFN-γ, and TNF-α production; however, intranasal instillation of EI137 and EI341 did not affect DP-induced IL-10 production by splenocytes (Figure 6).

### 3.6. Effect of Lidocaine Analogs on Histologic Changes in Sinonasal Mucosa

The DP challenge significantly increased inflammatory cell infiltration (1.8 ± 0.2) and goblet cell number (61.8 ± 5.5) compared to nonallergic negative control mice (0.4 ± 0.3 and 9.7 ± 6.7, respectively). Intranasal instillation of EI137 and EI341 at doses of both 0.5 and 5 μg/g significantly attenuated inflammatory cell infiltration (0.5 μg/g of EI137; 1.2 ± 0.3, 5 μg/g of EI137; 1.1 ± 0.1, 0.5 μg/g of EI347; 1.3 ± 0.2, and 5 μg/g of EI347; 1.2 ± 0.3) and goblet cell number (0.5 μg/g of EI137; 51.2 ± 11.8, 5 μg/g of EI137; 43.4 ± 7.2, 0.5 μg/g of EI347; 46.3 ± 25.9, and 5 μg/g of EI341; 32.3 ± 19.7) in the sinonasal mucosa (Figure 7A,B). Lidocaine also significantly reduced inflammatory cell infiltration (1.2 ± 0.3) and goblet cell number (43.9 ± 12.3); however, the increased epithelial thickness induced by DP stimulation was not affected by intranasal instillation of lidocaine, EI137, or EI341 (Figure 7C).

## 4. Discussion

Lidocaine is a local anesthetic agent with an immunomodulatory effect, inhibiting the production of inflammatory cytokines and modulating the immunologic function of inflammatory cells [14]. When nebulized, lidocaine exerts anti-inflammatory and anti-asthmatic effects, but it cannot be used clinically due to its local anesthetic effect and hyperstimulation of airway mucosa. Lidocaine-derived organic compounds, EI137 and EI341, demonstrated significant inhibition of eosinophil survival and activation induced by IL-5, as well as anti-inflammatory properties [10]. In the present study, we evaluated the effect of EI137 and EI341 on nasal allergic inflammation and investigated its possible underlying mechanism. The DP-induced AR mouse model exhibited more frequent nasal symptoms, increased infiltration of inflammatory cells, and inflammatory cytokine expression in the sinonasal mucosa. We found that the intranasal instillation of EI137 and EI341 significantly suppressed DP-induced upper airway allergic inflammation by inhibiting cytokine production and infiltration of sinonasal mucosal inflammatory cells. These anti-allergic and anti-inflammatory potencies were similar or even stronger than those exhibited by intranasal instillation of lidocaine.

AR is an IgE-mediated immediate type I inflammatory disease characterized by an imbalance between the Th1 and Th2 immune responses. DP is a major house dust mite allergen that induces an allergic response in sinonasal mucosa. Purified allergens from house dust mites, such as Der P1, and Der f1, have been shown to induce Th2-dominant allergic rhinitis in mice [15,16]. DP can also induce Th2 allergic responses with the production of Th2 cytokines and allergic symptoms [15]. In this study, intranasal instillation of DP induced both Th2 (IL-4) and Th1 (IFN-γ) responses in the serum and sinonasal mucosa. Crude house dust mite extract contains many allergens that can induce nonallergic inflammation by activating the complement system or other nonallergic innate and adaptive immune systems [17]. Intranasal instillation of DP significantly enhanced the expression of IFN-γ, IL-4, and IL-10 mRNA and GATA-3 in the sinonasal mucosa; however, although DP enhanced the expression of IFN-γ and IL-10 mRNA, it did not influence the mRNA expression of their transcription factors, T-bet and Foxp3. This may be attributed to the very low concentrations of IFN-γ and IL-10 in NLF and sinonasal mucosa [13]. Additionally, cytokines and their transcription factors may not always go together in target tissues. BALB/c mice have demonstrated both Th1 and Th2 immune-mediated airway inflammatory disease compared to other mice [16,18]. Intranasal EI137 and EI341 significantly inhibited the total IgE and Th2 cytokines in the serum and sinonasal mucosa, as well as IFN-γ and IL-10 mRNA expression in the sinonasal mucosa. To determine the anti-inflammatory effect of lidocaine analogs, cytokine production by mouse splenocytes was measured after stimulation with DP. Notably, DP significantly enhanced the production of IL-10, IFN-γ, IL-4, and TNF-α by splenocytes isolated from DP-challenged mice; however, the production of IFN-γ, IL-4, and TNF-α decreased in the splenocytes of mice that were intranasally treated with EI137 (5 μg/g) and EI341 (0.5 and 5 μg/g). This finding suggests that EI137 and EI341 can influence both Th1 and Th2 inflammatory processes; however, these lidocaine analogs did not influence IFN-γ levels, despite significantly suppressing IL-4 and TNF-α in the serum. Thus, intranasal administration of EI137 and EI341 may strongly inhibit Th1 and Th2 local inflammation while only affecting systemic Th2 inflammation. These anti-allergic and anti-inflammatory effects were more potent in higher concentrations (5 ug/g) of lidocaine analogs (IL-10, IL-4, and TNF-α in serum; IFN-γ, IL-4, and TNF-α in splenocytes). To determine the effect of the interaction between EI137 and EI341 on Th2 cytokines, we performed a docking study between lidocaine analogs and IL-4 and its receptor (Method S1). The docking results indicated that the lidocaine analog EI341 (EI137 is a positional isomer with EI341, so the interactions occur in the same way) exhibits a moderate binding affinity with a binding energy of -6.193 kcal/mol (Result S1 and Appendix A). This suggests that EI341 has the potential to modulate the IL-4–receptor interaction, demonstrating both biological activity and therapeutic potential.

Sneezing, rhinorrhea, and an itching sensation are characteristic clinical features of AR. Intranasal challenge with DP significantly increased sneezing, rubbing, and mucus-producing goblet cells in the sinonasal mucosa, all of which were significantly inhibited by intranasal instillation of EI137 and EI341. Eosinophils play an important role in the pathogenesis of an allergic reaction. Intranasal challenge with DP significantly enhanced inflammatory cell infiltration in the sinonasal mucosa as well as induced eosinophilia and neutrophilia in nasal secretions. This infiltration of inflammatory cells may be associated with an increase in TNF-α, a proinflammatory cytokine that promotes leukocyte infiltration and inflammation with stimulation of cytokine production [19]. Intranasal instillation of EI137 and EI341 suppressed DP-induced increases in serum TNF-α and its production in splenocytes. These lidocaine analogs significantly suppressed the infiltration of eosinophils (EI137; 66.9–91.1%, EI347; 38.9–81.5%) and neutrophils (EI137; 45.7–67.1%, EI347; 37.3–77.4%) in the sinonasal mucosa and nasal secretion; thus, lidocaine analogs can inhibit both eosinophilic and non-eosinophilic inflammation in the sinonasal mucosa. 

Our study has some limitations. First, the DP-induced AR mouse model and DP stimulation of splenocytes induced the production of both Th1 and Th2 cytokines; however, to determine the anti-allergic property of lidocaine analogs, an animal model with allergens specifically stimulating Th2 is ideal. Second, we did not determine the mechanism underlying the immunologic actions of these lidocaine analogs; this needs to be further evaluated in a mechanistic study. Third, the relatively small experimental sample size used in this study limited our analysis. To accurately understand the role of EI137 and EI341 on sinonasal inflammatory disease, further studies should utilize various concentrations of lidocaine analogs and other purified allergens, as well as a longer study period.

## 5. Conclusions

In this study, we assessed the anti-allergic and anti-inflammatory effects of lidocaine analogs, EI137 and EI341, on a house dust mite-induced AR mouse model. DP-challenges in sensitized BALB/c mice developed allergic symptoms, increased serum levels of IgE, IFN-γ, IL-4, IL-10, and TNF-α, and showed enhanced IFN-γ, IL-4, and TNF-α mRNA expression in the sinonasal mucosa. Intranasal instillation of EI137 and EI341 significantly influenced both Th1 and Th2 inflammatory cytokines and suppressed DP-induced sinonasal mucosal inflammation, inflammatory cell infiltration, and mucus hypersecretion. Based on these findings, the lidocaine analogs, EI137 and EI341, could be good candidates for topical anti-allergic and anti-inflammatory agents for upper airway inflammatory diseases.

## Figures and Tables

**Figure 1 biomedicines-12-01965-f001:**
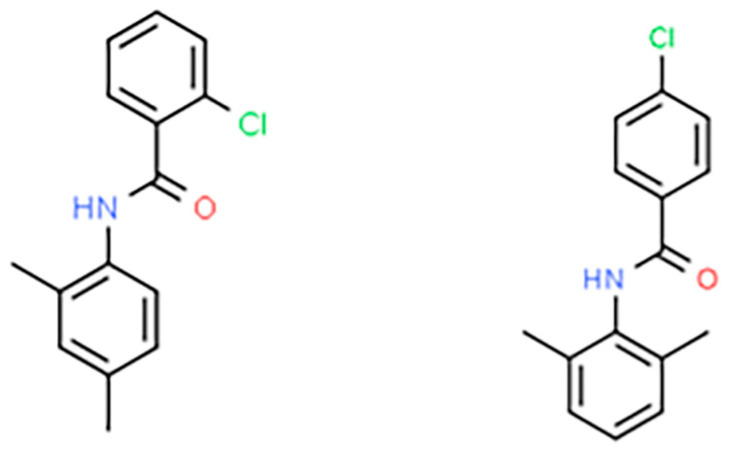
Chemical structures of EI137 and EI341.

**Figure 2 biomedicines-12-01965-f002:**
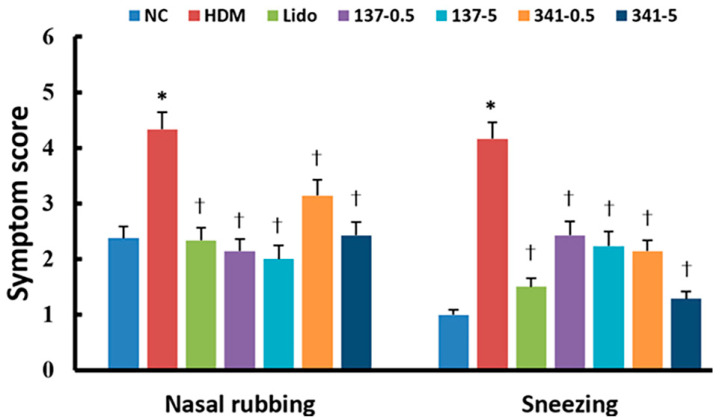
Effect of lidocaine-derived organic compounds, EI137 and EI341, on allergy symptoms in the house dust mite (HDM)-induced allergic rhinitis mouse model. The frequency of nasal rubbing and sneezing were significantly inhibited by 5 μg/g of lidocaine (Lido) and 0.5 μg/g and 5 μg/g of both EI137 (137–0.5, 137–5) and EI341 (341–0.5, 341–5). NC, negative control; *, *p* < 0.05 compared with NC; †, *p* < 0.05 compared with HDM (n = 8).

**Figure 3 biomedicines-12-01965-f003:**
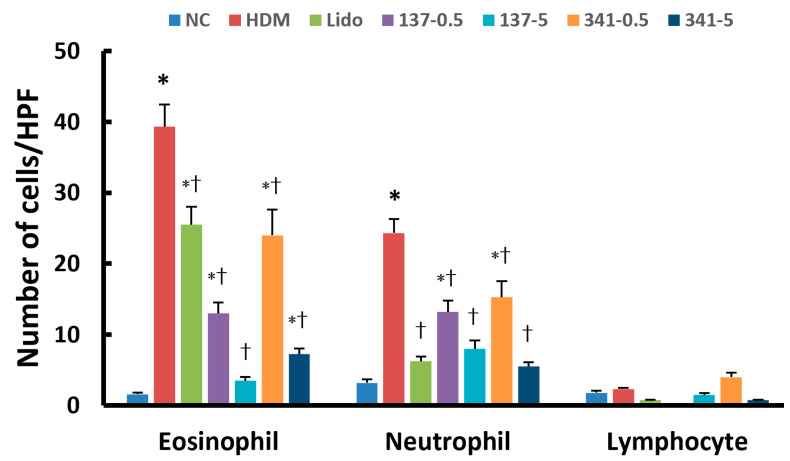
Effect of lidocaine-derived organic compounds, EI137 and EI341, on nasal lavage fluid inflammatory cell counts in house dust mite (HDM)-induced allergic rhinitis mouse model. Eosinophil and neutrophil counts were significantly decreased with intranasal instillation of 5 μg/g of lidocaine (Lido) and 0.5 and 5 μg/g of both EI137 (137–0.5, 137–5) and EI341 (341–0.5, 341–5). NC, negative control; *, *p* < 0.05 compared with NC; †, *p* < 0.05 compared with HDM (n = 8).

**Figure 4 biomedicines-12-01965-f004:**
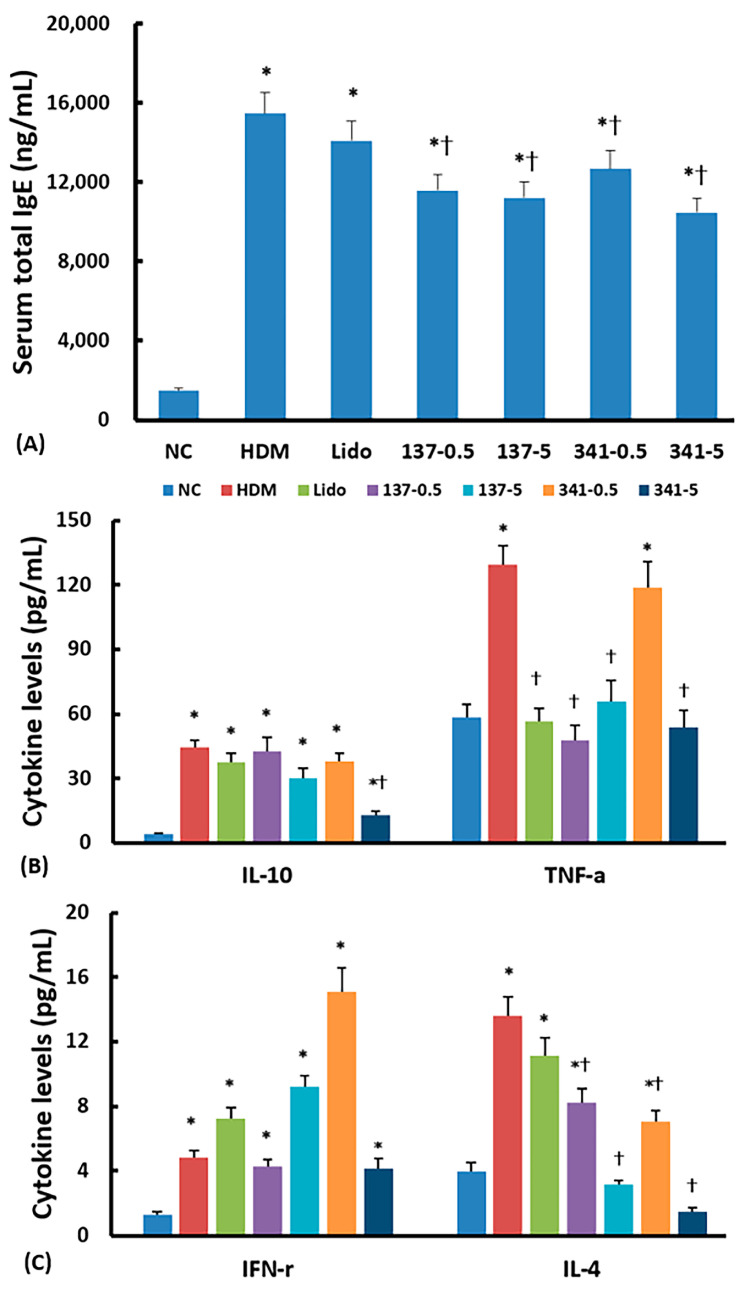
Effect of lidocaine-derived organic compounds, EI137 and EI341, on total serum IgE (**A**), IL-10 and TNF-α (**B**), and IL-4 and IFN-γ (**C**) levels in a house dust mite (HDM)-induced allergic rhinitis mouse model. A total of 0.5 μg/g and 5 μg/g of EI137 (137–0.5, 137–5) significantly suppressed IgE, IL-4, and TNF-α; 0.5 μg/g and 5 μg/g of EI341 (341–0.5, 341–5) significantly suppressed IgE and IL-4; and 5 μg/g of EI341 significantly suppressed IL-10 and TNF-α. NC, negative control; Lido, lidocaine; *, *p* < 0.05 compared with NC; †, *p* < 0.05 compared with HDM (n = 8).

**Figure 5 biomedicines-12-01965-f005:**
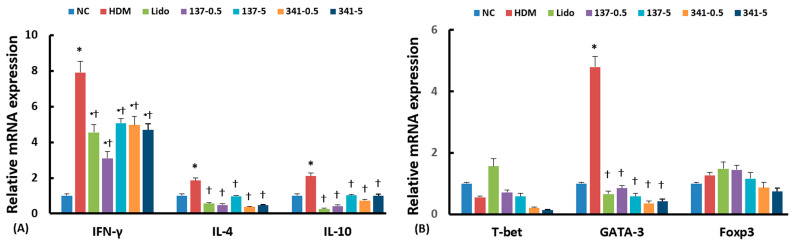
Effect of lidocaine-derived organic compounds, EI137 and EI341, on cytokine IFN-γ, IL-4, and IL-10 mRNA (**A**), and transcription factor T-bet, GATA-3, and Foxp3 mRNA (**B**) expression in the sinonasal mucosa in a house dust mite (HDM)-induced allergic rhinitis mouse model. The expressions of IFN-γ, IL-4, IL-10, T-bet, and GATA-3 mRNA were significantly inhibited by intranasal instillation of 5 μg/g of lidocaine (Lido) and both 0.5 and 5 μg/g of EI137 (137–0.5, 137–5) and EI341 (341–0.5, 341–5). NC, negative control; *, *p* < 0.05 compared with NC; †, *p* < 0.05 compared with HDM (n = 5).

**Figure 6 biomedicines-12-01965-f006:**
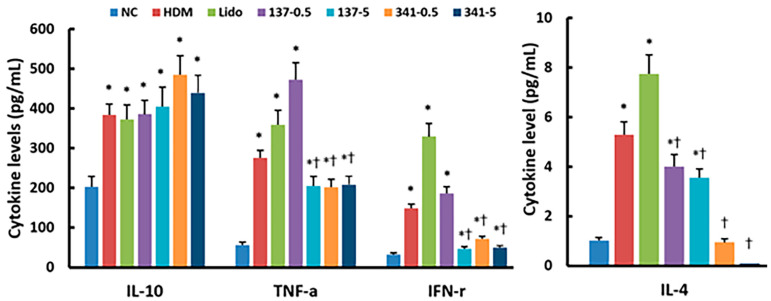
Effect of lidocaine-derived organic compounds, EI137 and EI341, on IL-10, TNF-α, IFN-γ, and IL-4 production by splenocytes in a house dust mite (HDM)-induced allergic rhinitis (AR) mouse model. Splenocytes from mice intranasally treated with 0.5 and 5 μg/g of EI341 (341–0.5 and 341–5), and 5 μg/g of EI137 (137–5) significantly decreased IL-4, IFN-γ, and TNF-α production versus nontreated AR mice. NC, negative control; *, *p* < 0.05 compared with NC; †, *p* < 0.05 compared with HDM (n = 8).

**Figure 7 biomedicines-12-01965-f007:**
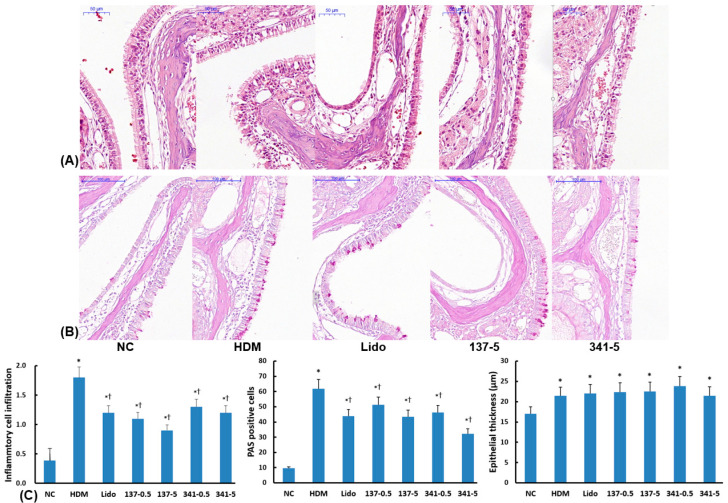
Effect of lidocaine-derived organic compounds, EI137 and EI341, on the histologic characteristics of the sinonasal mucosa in a house dust mite (HDM)-induced allergic rhinitis model. Inflammatory cell infiltration and periodic acid–Schiff (PAS)—positive cells were significantly decreased by intranasal instillation of 5 μg/g of lidocaine (Lido) and both doses (0.5 and 5 μg/g) of EI137 (137–0.5, 137–5) and EI341 (341–0.5, 341–5); however, epithelial thickness was not influenced by lidocaine and organic compounds (**C**). (**A**) Representative photographs of hematoxylin-and-eosin-stained tissues; (**B**) representative photographs of PAS-stained tissues. NC, negative control; *, *p* < 0.05 compared with NC; †, *p* < 0.05 compared with HDM (n = 5).

## Data Availability

Data are contained within the article and Appendix A, further inquiries can be directed to the corresponding author.

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
