# Peer review of "Anti-Allergic and Anti-Inflammatory Effects of Lidocaine-Derived Organic Compounds in a House Dust Mite-Induced Allergic Rhinitis Mouse Model"

_biomedicines, 2024, doi:10.3390/biomedicines12091965_

Round 1

Reviewer 1 Report

Comments and Suggestions for Authors

Comments:

This study evaluated the effect of lidocaine analogs in a Dermatophagoides pteronyssinus (DP)-induced AR mouse model, which is of some significance.

However, several parts of the text need to be revised, such as appropriately compressed preface; the content of organic compounds EI137 and EI341 should be provided; the grade and batch number of mice should be provided; the license number of the center supplying the mice animals should be provided; the ethical approval number should be provided if ethics has been passed; the specific method of modeling should be clarified, and the intraperitoneal sensitization of ovalbumin, which was mentioned in the abstract, has not been reflected in the main text; Part of the content under 2.8 is duplicated with that under 2.5, and should be reorganized and rearranged; part of the content is not logical, and should be checked, such as the title of 2.2, etc.; the manufacturers and batch numbers of the main reagents and medicines used for the test should be provided; the manufacturers and models of the main instrumentation used for the study should be provided; and there should be a method of drug administration and a basis for the dosage; The number of animals should be available at all the results and should be consistent with that at the methods; pathology pictures should be provided; references should be based on the literature of the last five years;[13]is repeated and [18] is missing in the main text.

It is recommended that the manuscript be withdrawn.

Comments on the Quality of English Language

Part of the content under 2.8 is duplicated with that under 2.5, and should be reorganized and rearranged; part of the content is not logical, and should be checked, such as the title of 2.2, etc.

Author Response

Thank you for taking the time to review our paper. We appreciate your valuable feedback and suggestions. We have carefully considered your comments and made the necessary revisions to improve the clarity and content of the manuscript.

This study evaluated the effect of lidocaine analogs in a Dermatophagoides pteronyssinus (DP)-induced AR mouse model, which is of some significance.

Comment 1) However, several parts of the text need to be revised, such as appropriately compressed preface;

Response) The introduction section has been re-written to improve its quality.

Comment 2) the content of organic compounds EI137 and EI341 should be provided;

Response) Molecular weight (259.7 g/mol) was described in Line 86 .

Comment 3) the grade and batch number of mice should be provided; the license number of the center supplying the mice animals should be provided;

Response) It is a supplier of laboratory animals licensed by the Ministry of Food and Drug Safety of the Republic of Korea (License No: 15). The animals are handled at the SPF grade level.
Unfortunately, the batch number cannot be confirmed.

Comment 4) the ethical approval number should be provided if ethics has been passed;

Response) Institutional Review Board Statement and approval number was provided. In Line 97 and Line 405 as ‘Institutional Review Board Statement: The animal study protocol was approved by the Institu-tional Review Board of Animal Experiments of Daegu Catholic University Medical Center (DCIAFCR-230213-39-YAA).

Comment 5) the specific method of modeling should be clarified, and the intraperitoneal sensitization of ovalbumin, which was mentioned in the abstract, has not been reflected in the main text;

Response) To clarify the AR modeling schedule, we added schematic diagram as Figure S1.
OVA was not used in the creation of the AR model, but it was mistakenly mentioned in the abstract. Abstract part (Line 19) was corrected.
Thank you for your careful comments.

Comment 6) Part of the content under 2.8 is duplicated with that under 2.5, and should be reorganized and rearranged;

Response) It seems that an error occurred while transferring the manuscript to the journal's provided template. The repeated text at the end of Section 2.8 has been removed.
Thank you once again for your careful point out.

 Comment 7) part of the content is not logical, and should be checked, such as the title of 2.2, etc.;

Response) Title of 2.2 was changes as ‘Development of DP-induced allergic rhinitis mouse model and experimental protocol’

Comment 8) the manufacturers and batch numbers of the main reagents and medicines used for the test should be provided; the manufacturers and models of the main instrumentation used for the study should be provided; and there should be a method of drug administration and a basis for the dosage;

Response) To provide information about the reagents used, Section 2.1 has been added.
The model names of the equipment used was added.

 Comment 9) The number of animals should be available at all the results and should be consistent with that at the methods;

Response) Each experimental group consisted of 8 mice. Histologic and mRNA studies used 5 mice each, while the remaining studies on inflammatory cells and cytokines were conducted with 8 mice per group.
To clarify, animal number was added in legend of each figure.

Comment 10) pathology pictures should be provided;

Response) H&E and PAS stained images were added in Fig 7, as recommended.

Comment 11) references should be based on the literature of the last five years;[13]is repeated and [18] is missing in the main text.

Response) There is limited recent research on the anti-inflammatory and anti-allergic effects of lidocaine, and the introduction includes a few studies that provide the basis for the research, so not all references could be from the past 5 years. However, some of the references have been updated to more recent ones.
Other issues related to the references have also been corrected and revised (Ref 8, 17, and 18 were changed).

I hope the revised manuscript will better meet the requirements for publication.

Thank you

Reviewer 2 Report

Comments and Suggestions for Authors

Dear Authors,

I appreciate the opportunity to review your manuscript on the effects of lidocaine analogs EI137 and EI341 in a mouse model of allergic rhinitis. The study presents interesting findings, but I have several suggestions that could enhance the clarity and depth of your work:

1. The method used to calculate the symptom score in Figure 2 is not clearly explained in the manuscript. Please provide a detailed description of how this score was derived.

2. I recommend including images of the sinonasal mucosa tissue sections stained for histological evaluation in the allergic rhinitis model. This would provide visual evidence to support your findings.

3. It would be beneficial to include an existing allergic rhinitis treatment as a positive control in your experiments. This comparison could strengthen the validity of your results.

4. Further investigation into the mechanisms of action of the lidocaine analogs EI137 and EI341 would be valuable. Consider conducting pathway studies to elucidate their effects on inflammatory processes.

5. I suggest testing a wider range of concentrations for EI137 and EI341. This could help determine the dose-response relationship and optimize their therapeutic potential.

6. Please clarify the rationale behind the selection of the concentrations used for EI137 and EI341 in your experiments. This information is crucial for understanding the experimental design.

7. The results and images related to the docking studies of the lidocaine analogs with IL-4 and its receptor seem to be missing from the manuscript. Including these findings would provide additional insights into the interactions and potential mechanisms of action.

Thank you for considering these suggestions. I believe that addressing these points will significantly improve the quality of your manuscript. I look forward to seeing the revised version.

Best regards,

Comments on the Quality of English Language

Minor editing of English language required

Author Response

Thank you for taking the time to review our paper. We appreciate your valuable feedback and suggestions. We have carefully considered your comments and made the necessary revisions to improve the clarity and content of the manuscript.

Comments 1). The method used to calculate the symptom score in Figure 2 is not clearly explained in the manuscript. Please provide a detailed description of how this score was derived.

 Response) To clarify the evaluation of allergic symptoms, section 2.3 was re-written as ‘Mice were placed into an observation cage for approximately 10 min for acclimatization. Then, 10 μg of DP was administrated intranasally, and nasal symptoms were evaluated 5 min later by counting the time of nasal rubbing and the number of sneezing over a 15 min period following the last stimulation.’ In Line 111-114.

Comments 2). I recommend including images of the sinonasal mucosa tissue sections stained for histological evaluation in the allergic rhinitis model. This would provide visual evidence to support your findings.

 Response) H&E and PAS stained images were added in Fig 7, as recommended.

Comments 3). It would be beneficial to include an existing allergic rhinitis treatment as a positive control in your experiments. This comparison could strengthen the validity of your results.

 Response) Answer) It is known that intranasal administration of lidocaine in a murine allergic rhinitis model improves allergic inflammation and allergic symptoms (Ref 12; Exp Ther Med 2022 Mar;23(3):193). Additionally, lidocaine inhalation has been reported to suppress bronchial mucosal inflammation (Int Immunopharmacol 2008 May;8(5):725-31).

Based on these references, the authors used lidocaine as a positive control in this study.

And it was mention in line 107-9, as ‘Intranasal instillation of lidocaine is known to have anti-inflammatory and anti-allergic effects in a murine model of allergic rhinitis, we used lidocaine as a positive control [12].’

Comments 4). Further investigation into the mechanisms of action of the lidocaine analogs EI137 and EI341 would be valuable. Consider conducting pathway studies to elucidate their effects on inflammatory processes.

 Response) As you pointed out, it is important to investigate the anti-inflammatory effects of lidocaine analogs and their mechanisms. However, this study focused on the anti-allergic effects of lidocaine analogs. Therefore, the experiments were focused on Th1 and Th2-related cytokines and transcription factors.

Comments 5}. I suggest testing a wider range of concentrations for EI137 and EI341. This could help determine the dose-response relationship and optimize their therapeutic potential.

 Response) In vitro studies using EI137 and EI341 were conducted at various concentrations. However, in vivo studies had several limitations, so we conducted the study at two concentrations and conducted the experiment on a total of 7 groups.
If additional study with other concentrations are needed, we will conduct additional experiments upon your opinion.

Comments 6). Please clarify the rationale behind the selection of the concentrations used for EI137 and EI341 in your experiments. This information is crucial for understanding the experimental design.

 Response) In a previous in vitro study comparing lidocaine, EI137, and EI341 (Ref 10; Molecules 2023, 28, 5696), they exhibited similar anti-inflammatory effects at the same concentrations.
We determined experimental concentrations based on a previous study (Ref 12; Exp Ther Med 2022 Mar;23(3):193), which demonstrated that 5 mg/kg of lidocaine had an anti-inflammatory effect in an allergic rhinitis murine model.
Since lidocaine and lidocaine analogs showed similar anti-inflammatory effects at comparable concentrations, we conducted our experiments using 5 mg/kg lidocaine and 0.5 mg/kg and 5 mg/kg EI137 and EI341.

Commetns 7). The results and images related to the docking studies of the lidocaine analogs with IL-4 and its receptor seem to be missing from the manuscript. Including these findings would provide additional insights into the interactions and potential mechanisms of action.

Response) We apologize for not notifying you that the results for the docking study have not been submitted. The method and results have been submitted as supplementary data.

 I hope the revised manuscript will better meet the requirements for publication.

Thank you

Reviewer 3 Report

Comments and Suggestions for Authors

Why only two analogs were selected?

Why a positive control is not used? It would have helped in understanding the effectiveness of the analogs.

Figure 3, also includes lymphocytes and other cells. Though their impact is not prominent yet they need to be described in the results. Also describe what the term “others” include.

Page 6. Line 221 “ecreased by intranasal instillation of EI137 and EI137 a” please re check the sentence.

Page 6, Line 222 “However, IgE levels were not significantly affected by intranasal instillation of 5 μg/g of lidocaine (Figure 4A)” the authors claim non-significant affect while in figure there is an asterisk (*) with lido.

Figure 4: Proper symbols need to be inserted. (ϒ,α)

In figure 6: The bars in IFN, visually there seems to be less difference between NC and 137-5,341-0.5, and 341-5, however, an asterisk is placed. How can it be justified?

Page 9, line 275, “compared to nonallergic negative control mice (0.4 ± 0.3)” is it the value in figure 7 (A)? 0.3 the standard deviation is a little high and the standard error bar is not reflecting 0.3. Please recheck all the figures for mean ± SD value.

The supplementary data cannot be accessed through the link provided.

Comments on the Quality of English Language

Minor changes required. 

Author Response

Thank you for taking the time to review our paper. We appreciate your valuable feedback and suggestions. We have carefully considered your comments and made the necessary revisions to improve the clarity and content of the manuscript.

Comments 1) Why only two analogs were selected?

Response) Previous, we conducted a study on the activation and survival of eosinophils using 36 lidocaine derived organic compounds. Among them, authors identified that two compounds, EI137 and EI341, had inhibitory effects on eosinophil activation and survival (Ref 10; Molecules 2023, 28, 5696).
Based on these findings, we studied using EI137 and EI314 in this study.

And it was mention in line 66, as ‘among 36 lidocaine-derived……’

Comments 2) Why a positive control is not used? It would have helped in understanding the effectiveness of the analogs.

Response) It is known that intranasal administration of lidocaine in a murine allergic rhinitis model improves allergic inflammation and allergic symptoms (Ref 12; Exp Ther Med 2022 Mar;23(3):193). Additionally, lidocaine inhalation has been reported to suppress bronchial mucosal inflammation (Int Immunopharmacol 2008 May;8(5):725-31).
Based on these references, the authors used lidocaine as a positive control in this study.

And it was mention in line 107-9, as ‘Intranasal instillation of lidocaine is known to have anti-inflammatory and anti-allergic effects in a murine model of allergic rhinitis, we used lidocaine as a positive control [12].’

Comments 3) Figure 3, also includes lymphocytes and other cells. Though their impact is not prominent yet they need to be described in the results. Also describe what the term “others” include.

Response) "Others" was used to express a few unspecified cells, such as mast cells. However, to avoid confusion for the readers, "Others" was removed from Figure 3.
And In line 211-3, we added information regarding lymphocytes, as ‘Although a few lymphocytes were observed in the NLF, there was no difference between the experimental groups.’

Comments 4) Page 6. Line 221 “ecreased by intranasal instillation of EI137 and EI137 a” please re check the sentence.

Response) EI137 and EI341 suppressed DP induced IgE production. To clarify in line225, ‘that were increased by DP’ was added.

Comments 5) Page 6, Line 222 “However, IgE levels were not significantly affected by intranasal instillation of 5 μg/g of lidocaine (Figure 4A)” the authors claim non-significant affect while in figure there is an asterisk (*) with lido.

Response) As described in legend, * means significantly increased in compare to negative control group and † means significantly decreased in compare to HDM group.

The serum IgE levels of all experimental mice were significantly increased compared to the negative control mice. While the DP-induced increase in serum IgE levels was not affected by lidocaine, it was reduced by EI137 and EI341.

Comments 6) Figure 4: Proper symbols need to be inserted. (ϒ,α)

Response) I was unable to make the corrections as I couldn't identify the issue you mentioned.
If you could clarify the point again, I will correct as recommend. Thank you.

Comments 7) In figure 6: The bars in IFN, visually there seems to be less difference between NC and 137-5,341-0.5, and 341-5, however, an asterisk is placed. How can it be justified?

Response) The amount of IFN-γ was relatively low compare to IL-10 and TNF-α. These three cytokines were represented in a single graph, which may have caused some confusion.
The level in the negative control was 39.1±11.3 pg/mL, while it was 46.9±13.4 pg/mL in EI137-5, 71.4±24.7 pg/mL in EI341-0.5, and 49.9±21.1 pg/mL in EI341-5. The IFN-γ levels were significantly higher in these three experimental groups compared to the negative control group.

Comments 8) Page 9, line 275, “compared to nonallergic negative control mice (0.4 ± 0.3)” is it the value in figure 7 (A)? 0.3 the standard deviation is a little high and the standard error bar is not reflecting 0.3. Please recheck all the figures for mean ± SD value.

Response) Thank you for your careful observation. I have checked and corrected it in Figure 7.

Comments 9) The supplementary data cannot be accessed through the link provided.

Response) We apologize for not notifying you that the results for the docking study have not been submitted. The method and results have been submitted as supplementary data.  

I hope the revised manuscript will better meet the requirements for publication.

Thank you.

Round 2

Reviewer 1 Report

Comments and Suggestions for Authors

The authors of this article have addressed or provided explanations for the issues raised, but there are still some questions, as follows: 

First, the content and quality control standards of EI137 and EI34 should be given. In addition, the qualification number of this batch of mice should be provided, as well as the license number of the animal center that supplied the animals. Finally, why is the sample number N = 5 in both Figures 5 and 7?

In view of the above factors, the content of this study does not meet the journal's requirements for the publication of the article, and it is recommended that the article be rejected.

Reviewer 2 Report

Comments and Suggestions for Authors

Dear Authors,

I would like to commend you on the revisions made to your manuscript in response to the comments provided during the first round of peer review. It is evident that you have taken the feedback seriously and have made significant improvements to the clarity and quality of your work.

The modifications you have implemented effectively address the concerns raised in the initial review. The changes enhance the overall coherence of the manuscript and provide a more robust presentation of your findings.

I believe that your manuscript is now in a much stronger position for publication. I commend your efforts in revising the manuscript and look forward to seeing it published.

Thank you for your hard work and dedication to this research.

Best regards,

Reviewer 3 Report

Comments and Suggestions for Authors

The comment regarding figure 6 was to use appropriate symbol on the horizontal x-axis. Such corrections can be made during proof-reading.